# Association between Participation Activities, Pain Severity, and Psychological Distress in Old Age: A Population-Based Study of Swedish Older Adults

**DOI:** 10.3390/ijerph18062795

**Published:** 2021-03-10

**Authors:** Elena Dragioti, Björn Gerdle, Lars-Åke Levin, Lars Bernfort, Huan-Ji Dong

**Affiliations:** 1Pain and Rehabilitation Centre, Department of Health, Medicine and Caring Sciences, Linköping University, SE-581 85 Linköping, Sweden; elena.dragioti@liu.se (E.D.); bjorn.gerdle@liu.se (B.G.); 2Department of Health, Medicine and Caring Sciences, Division of Health Care Analysis, Linköping University, SE-581 85 Linköping, Sweden; lars-ake.levin@liu.se (L.-Å.L.); lars.bernfort@liu.se (L.B.)

**Keywords:** participation, older adults, leisure activity, psychological distress, gender, body mass index

## Abstract

Although chronic pain is common in old age, previous studies on participation activities in old age seldom consider pain aspects and its related consequences. This study analyses associations between participation activities, pain severity, and psychological distress in an aging population of Swedish older adults (*N* = 6611). We examined older adults’ participation in five common leisure activities using the Multidimensional Pain Inventory (MPI), sociodemographic factors, pain severity, weight status, comorbidities, and pain-related psychological distress (anxiety, depression, insomnia severity, and pain catastrophising). We found that gender, body mass index (BMI) levels, and psychological distress factors significantly affected older adults’ participation in leisure activities. Pain severity and multimorbidity were not significantly associated with older adults’ participation in leisure activities nor with gender stratification in generalised linear regression models. The potentially modifiable factors, such as high levels of BMI and psychological distress, affected activity participation in men and women differently. Health professionals and social workers should consider gender and target potentially modifiable factors such as weight status and psychological distress to increase older adults’ participation in leisure activities.

## 1. Introduction

Today, participation has become a significant topic when discussing health-related outcomes such as well-being, life satisfaction, and happiness [1,2,3,4]. Moreover, the research on the benefits of increased well-being and life satisfaction is receiving attention from the health care system [5,6,7]. Although there is not a widely accepted definition, participation in leisure activities can be defined as the use of relaxing, interesting, and enjoyable activities spent outside of work that fosters well-being and life satisfaction [3]. Research on this topic has shown a positive correlation between positive health-related outcomes and different aspects of leisure activities such as playing sports, games, and cards; visiting friends; making excursions; and taking trips [1,2,4,8]. Within this broad framework, the concept of participation in leisure activities is one of the most critical issues among the elderly population and goes with the concept of “successful aging” [9,10]. Successful aging is defined by the World Health Organization (WHO) as “the process of optimizing opportunities for health, participation and security to enhance quality of life as people age” [11].

A large body of work has also documented that participation spheres such as outdoor activities, social life, leisure activity, and physical activity positively influence several aspects of elderly’s health [2,4,9,12,13,14,15,16]. Specifically, elderly who are active have increased happiness, enhanced life satisfaction and health-related quality of life, better mental health [10,14,16], increased endurance and flexibility [17], decreased levels of inflammatory biomarkers [18], and reduced cardiovascular risk [19,20]. Moreover, participation in leisure activities offers an array of positive feelings for the elderly (e.g., autonomy and independency, fulfilment, joy, interest, and hope) that help them create a balance in their life, cope with daily life stress, and maintain social interactions [17,21,22].

On the contrary, a reduction in participation in leisure activities as people age has been associated with negative outcomes. Reduced participation can lead to lack of intimate relationships, dependency, and poor health—outcomes that increase feelings of loneliness, which can contribute to greater disability [10]. For example, Orsega-Smith et al. found that older adults who participated in a community-based exercise program more frequently gained better physical, psychological, and social benefits than those who did not participate [17]. Another study found that participation in gender-diverse groups is linked to a better health profile [23]. However, older adults are often debilitated because of poor physical and mental health, which may affect their participation rate in leisure activities [10], resulting in unhealthy sedentary behaviour [6].

With the population aging, the number of older adults experiencing reduced leisure participation will increase. About 60% of adults aged 65 years and over experience reduced participation rates, and after the age of 90 more than 30% of the older adults are socially inactive [24]. More importantly, the variation of participation in the elderly seems to depend on their physical and psychological health condition [10]. For many diseases, pain and psychological distress (e.g., insomnia, depression, and anxiety) interfere with participation in leisure activities in older adults [25,26,27,28]. Recent data have shown that participation in leisure activities reduces the risk of developing chronic pain, and participation in physical activities can be considered as a non-pharmacological approach to pain management [29,30]. However, these studies either focused on physical activity in general or neglected the gender role in participation activities related to chronic pain. Given the complex nature of aging, physical health, psychological health [10], and gender should be considered when investigating participation aspects among the elderly [31,32]. To this end, this study examines the participation in the most common leisure activities in older adults in relation to pain and psychological distress as well as gender, since engaging in leisure activities plays an important role in successful aging.

## 2. Materials and Methods

### 2.1. Study Population

This is a large cross-sectional study based on the Swedish Total Population Register (STPR) for the two largest cities (Linköping and Norrköping) in one county (Östergötland) in south-eastern Sweden [33,34,35,36,37]. The STPR for this geographical area entails about 49,320 persons in the age group 65 years and older. A stratified sample of 2000 subjects based on five age strata (65 to 69 years, 70 to 74 years, 75 to 79 years, 80 to 84 years, and 85 years and older) was randomly selected from the STPR. Hence, a total of 10,000 subjects were eligible to be included. A postal survey of pain intensity, anxiety, depression, pain catastrophising, insomnia, and sociodemographic characteristics (age, gender, and education level) was mailed in October 2012 and closed in January 2013. We classified the educational level into the following three categories: compulsory school (elementary/secondary), upper secondary school (or vocational training), and university/college. If necessary, the postal survey was followed by as many as two postal reminders (at two-week intervals). Data were collected by Statistics Sweden (SCB), which also provided information about marital status and yearly income (Swedish Crowns 0–150,000, 150,001–220,000, and 220,001 and over) from the Swedish Population Register [33]. The study was approved by the Regional Ethics Research Committee in Linköping, Sweden (Dnr: 2012/154-31) on 18 June 2012.

### 2.2. Measurements

#### 2.2.1. Leisure Activities

Leisure activities were measured using the Multidimensional Pain Inventory (MPI). The MPI is a 61-item self-report questionnaire in which recipients are asked to respond on a seven-point numerical scale (range 0–6). The MPI is divided into one psychosocial (Part 1) and two behavioural (Parts 2 and 3) sections, making a total of nine empirically derived scales [38,39]. We chose five scales (playing cards or other games, visiting friends, taking a ride in a car, visiting relatives, and making an excursion) concerning leisure activities from Part 3 (19 items), which comprises a list of common activities that patients rate in terms of the frequency with which they perform each activity, according to the Swedish version of MPI (MPI-S) [40]. These subscales were summarised and divided by number of items answered (not all subjects answered items concerning car and relatives within 150 km) in order to form a leisure index (MPI-leisure total score) according to the validation of the Swedish version of MPI-S. The instrument has provided good psychometric properties [38,39]. The test-retest intraclass coefficient of MPI-leisure activities was 0.79 in the MPI-S [40].

#### 2.2.2. Pain Intensity and Pain Severity

Pain intensity was measured by an eleven grade (0–10) numeric rating scale: zero indicated no pain at all and 10 indicated worst possible pain [41]. For analytical purposes [33], the population was subdivided into three groups with respect to chronic pain and pain severity during the previous week: no or mild pain (0–4), moderate pain (5–7), and severe pain (8–10).

#### 2.2.3. Anxiety and Depression

Two subscales of the General Well-being Schedule (GWBS) were used to assess anxiety and depression. The GWBS, a common instrument for assessing life satisfaction and level of psychological distress [42], has six subscales with 18 items and has good internal consistency, test–retest reliability, and validity [42]. The subscale GWBS-Anxiety consists of items 2, 5, 8, and 16, ranging between 0 and 25. The subscale GWBS-Depression consists of items 4, 12, and 18, ranging between 0 and 20. These items were reverse-scored so high values indicate higher symptoms of anxiety and depression [35].

#### 2.2.4. Pain Catastrophising

The Pain Catastrophizing Scale (PCS) measures three dimensions of pain catastrophising—rumination, magnification, and helplessness—based on 13 items each with five alternatives: 0 = “not at all”; 1 = “to a slight degree”; 2 = “to a moderate degree”; 3 = “to a great degree”; and 4 = “all the time” [43]. It has adequate to excellent internal consistency and validity [44,45]. Here, we used the total PCS (PCS-total): the maximum score according to the original scale was 52 with a high score indicating a worse outcome. However, due to a printing malfunction, the most negative alternative (“all the time”) was not printed in the questionnaire, so the most negative alternative was “to a great degree” [35,37]. In this study, PCS-total had possible scores between 0 and 39 instead of 0 and 52. The internal consistency of the PCS-total in this study was α = 0.75.

#### 2.2.5. Insomnia Severity

Insomnia was assessed with the Insomnia Severity Index (ISI). The ISI is a seven-item self-report instrument assessing the nature, severity, and impact of insomnia [46,47]. A five-point Likert scale is used to rate each item (0 = no problem and 4 = very severe problem), yielding a total score ranging from 0 to 28. The total score is divided into four categories: no clinically significant insomnia (ISI: 0–7); sub-threshold insomnia (ISI: 8–14); moderate clinical insomnia (ISI: 15–21); and severe clinical insomnia (ISI: 22–28). ISI is a reliable and valid instrument to detect cases of insomnia with excellent internal consistency [47]. This study presents results only for the total score.

#### 2.2.6. Comorbidities and Multimorbidity

The evaluation of comorbidities was based on a 12-item self-reported list covering different aspects of common comorbidities: (1) traumatic accident, (2) rheumatic arthritis and osteoarthritis, (3) cardiovascular diseases (CVD; including high blood pressure, angina pectoris, and heart attacks), (4) diseases of airways or lungs, (5) low mood and depression, (6) anxiety, (7) diseases of the gastrointestinal system, (8) diseases of the nervous system, including eyes and ears, (9) diseases of the urogenital organs, (10) diseases of the skin, (11) tumours and cancer, and (12) metabolic diseases (including diabetes, obesity, anorexia, bulimia, and goitre). These comorbidities were reported on a five-graded scale: (1) no; (2) yes, according to both my own and my doctor’s opinions; (3) yes, according to my own opinion; (4) yes, according to my doctor’s opinion; and (5) do not know. We combined the answers from 2 and 4 to have a robust measurement of the presence of a certain comorbidity [36]. These items were dichotomised as follows: “yes, according to both my own and my doctor’s opinions plus according to my doctor’s opinion” versus the three other alternatives. Based on these dummy variables in each comorbidity, a total index score was calculated as the total number of comorbidities for analytical purposes. In accordance with the literature, multimorbidity was defined as two or more comorbidities [48,49,50].

### 2.3. Statistical Analysis

The statistics were performed using the statistical package IBM SPSS Statistics (version 25.0; IBM Inc., New York, NY, USA). Level of significance was set to <0.05 in all tests. The normality of the scores based on Skewness and Kurtosis of all measures was investigated and found to be within acceptable values for normality (≥±2.00 for skewness and ≥±7.00 for kurtosis) [51]. Distributions and descriptive statistics were examined for all variables. Continuous data are reported as the mean and standard deviation (SD) and the categorical data are represented as *n* (%). The independent Student’s t-test and Chi-square test were used to examine links between gender and other variables. Pearson’s r was used to examine possible links between participation in leisure activities and other factors (i.e., socio-demographics, pain severity, BMI, number of comorbidities, and psychological distress variables).

Generalised linear regression analyses (maximum likelihood) were performed to understand the importance of gender, pain severity, psychological distress (GWBS-depression, GWBS-anxiety, pain catastrophizing, and insomnia), and multimorbidity (exploratory variables) to predict leisure activities (MPI-leisure; dependent variable), after adjusting for other sociodemographic factors. The estimated effects were presented as parametric estimates (B) with 95% confidence interval (CI) using Wald statistic. A high collinearity was found between GWBS-depression and GWBS-anxiety (Pearson’s r = 0.75). GWBS-depression was included in the models instead of GWBS-anxiety due to fewer missing cases (383 vs. 419). Finally, we performed generalised linear regression analyses with a gender stratification.

## 3. Results

### 3.1. Description of General Characteristics

The final sample consisted of 6611 respondents (response rate: 66.5%). More than 1100 responses (11.5%) were received after two postal reminders. Details of socio-demographic characteristics and all examined variables of the sample are presented in Table 1. There were 3057 men (46.2%) and 3554 women (53.8%). The average age of the total sample was 76.2 years (range, 65–102 years). The majority (57%) of respondents, more men (68.9%) than women (46.8%), were currently married (*p* < 0.001). Approximately every other respondent had lowest education level (compulsory school, 52.3%). A lower proportion of women (20.1%) than men (23.8%) earned university or college education (*p* < 0.001). Over one in three participants (35.9%) were classified into the highest income level (>220,000 Swedish Crowns/year, 2012). More than half of men (53.6%), whereas only one-fifth of women (20.7%), had the highest income level (*p* < 0.001). Most women’s incomes (47.5%) were classified into the lowest level (<150,000 Swedish Crowns/year), and there was only a minority of men (9.4%) in this level (*p* < 0.001).

Most respondents reported no pain or mild pain (77%), with a slightly higher proportion of men (82%) than women (73%) (*p <* 0.001). Over a half of the sample was overweight (39%) or obese (14%). More women than men were normal weight (47% vs. 43%) (*p* < 0.001). There were more men than women in the overweight category (43% vs. 35%). A slightly higher proportion of women than men were obese (15% vs. 13.5%). Women scored significantly higher levels in all four psychological distress aspects (*p* < 0.01~0.001). Multimorbidity existed in 53% of the whole sample, with a slightly higher proportion of women (55%) than men (51%) (*p <* 0.001).

The mean MPI-leisure total score was 2.4 (SD 1.1). The mean MPI-leisure total score for men was 2.4 (SD 1.0) and for women was 2.5 (SD 1.1) (*p* < 0.001). Men scored higher than women in only one activity—taking a ride in a car (1.7 vs. 1.5) (*p* < 0.001). However, almost 30% of the women (*n* = 1038) did not have a car and about 10% of the men did not have a car (*n* = 336).

### 3.2. Variables in Correlation with Leisure Activities Participation

Sociodemographic factors including age, currently married, and income level had significant but low correlations with MPI-leisure total score (Table 2). MPI-leisure total score was negatively and significantly correlated with pain intensity, number of comorbidities, and psychological distress variables including depression, anxiety, insomnia, and catastrophising. The gender differences were found in the three correlations: MPI leisure score had no significant correlations with education level, pain intensity, and catastrophising in men, but they were positively or negatively correlated in women.

### 3.3. Predictors of Participation in Leisure Activities in Older Adults

Table 3 presents the results of multivariable regression models testing the linear links between participating in leisure activities and other potential predictors. In the whole sample population, female gender was positively associated with the MPI-leisure total score (B = 0.25, 95% CI 0.14, 0.37) (*p <* 0.01) and three social activities—visiting friends, visiting relatives, and making an excursion (*p <* 0.01~0.001). Compared to being normal weight, being overweight was positively associated with MPI-leisure total score (B = 0.22, 95% CI 0.11, 0.33) (*p <* 0.001) as well as positively associated with all the five activities (*p <* 0.05~0.001). Being obese, however, increased the likelihood to play cards/other games and take a ride in a car (*p <* 0.05), but being obese meant one was less likely to go on an excursion (*p <* 0.05). Depression was negatively and strongly associated both with MPI-leisure total score (B = −0.61, 95% CI −0.81, −0.40) (*p <* 0.001) and with participation in all five activities (*p <* 0.01~0.001). Severity of insomnia was negatively linked to MPI-leisure total score (B = −0.02, 95% CI −0.03, −0.01) (*p <* 0.01) and participation in three activities (*p <* 0.05~0.001). Pain catastrophising was only weakly related to participation in making an excursion (B = −0.01, 95% CI −0.02, −0.01) (*p <* 0.05). Neither pain levels nor multimorbidity showed significant association with participation in leisure activities.

As shown in Table 4, the regression models stratified gender. Several differences were found between older men and women with respect to participation in leisure activities. For pain levels and multimorbidity, the only significant, although weak, effect was found in women with severe pain who participated in visiting friends (B = 0.49, 95% CI 0.09, 0.89) (*p <* 0.05). Among the older men, being overweight was positively associated with playing cards/other games (*p <* 0.05) and taking a ride in a car (*p <* 0.001). In women, the positive associations were linked to visiting friends and visiting relatives (*p <* 0.05). High BMI, although below the obesity level, was negatively associated with men’s participation in making an excursion (B = −0.40, 95% CI −0.74, −0.05) (*p <* 0.05), but it did not affect women significantly (*p* > 0.05).

Variables on psychological distress also affected men and women differently. Depression was more strongly and negatively related to all women’s activity participation (*p <* 0.01~0.001) except taking a ride in a car, which was negatively associated with men’s depression (B = −0.72, 95% CI −1.30, −0.15) (*p <* 0.05). Insomnia negatively affected both men’s and women’s participation in visiting friends (*p <* 0.01~0.001), men’s participation in visiting relatives (*p <* 0.05), women’s participation in making an excursion (*p <* 0.001), as well as women’s MPI-leisure total score (*p <* 0.05). Pain catastrophising, however, only contributed negatively to women’s friends visiting (B = −0.02, 95% CI −0.03, −0.01) (*p <* 0.01) and did not affect any men’s activity participation.

## 4. Discussion

After adjusting for other sociodemographic factors, we found that gender, BMI status, and psychological distress factors were linked to participation in leisure activities in old age. Surprisingly, pain severity and multimorbidity were neither associated with MPI-leisure as a total score nor with any of the five activities. With respect to gender, the influences of pain severity and multimorbidity only negligibly affected participation in most leisure activities. The potentially modifiable factors, such as high levels of BMI and psychological distress, affected men and women differently in their activity participations. Thus, our results suggest that prevention and intervention approaches to these potentially modifiable factors should improve activity participation in older adults. One strength of this study is its large-scale cross-sectional analysis with random samples by age strata as well as gender stratification. The large sample size enables us to use sample stratifying while maintaining adequate statistical power for data interpretations.

The five participation activities from MPI-S cover the physical, social, and cognitive aspects based on a cognitive-behavioural perspective [40,52]. This questionnaire was developed for pain-related consequences and coping strategies, which were applicable in our research on investigating the elderly population with respect to pain characteristics and how pain was correlated to comorbidities, activities, and health [33]. Overall, we found MPI-leisure total score was not highly correlated to any single variable examined in the whole sample. This finding indicates an underlying complex relationship between these factors and activity participation in this heterogenous study population. In the regression analysis, female gender was positively associated with MPI-leisure total score as well as two activities in connection with friends or relatives and one activity that was both physically and cognitively demanding (i.e., going on an excursion). This gender gap is not in agreement with previous studies that report men spend more time in leisure activities than women [53,54]. Since there is no gold standard explanation, we assume that in Sweden compared to other countries women of that age spend fewer hours doing house work [55] and are more innovative in their leisure in older age [54]. On the other hand, this gender distinction is in line with the findings from sociological research that women have better communicative skills than men with respect to maintaining connections with friends and other family members [56].

Explaining the role of gender in physical- and cognitive-related activities is more complex as other factors need to be considered. Together with gender, positive associations were found between being overweight and participating in all activities. Obesity’s positive associations to participation in sedentary activities (more likely to play cards/other games and take a ride in a car) and negative associations to participation in physically demanding activities (less likely to go an excursion) suggest that physical inactivity/sedentary lifestyle is linked to obesity. A large body of research clarifies the relationship between physical activity, physical functioning, physical health, and excess weight [57,58,59,60]. To the best of our knowledge, benefits from being overweight (opposed to obesity) in activity participation have not been previously reported in the literature. One explanation is that these leisure activities were not only limited to physical functioning but also related to other functions such as cognitive function. The relationship between cognitive function and BMI was controversial in earlier studies [61,62,63]. Another speculation is that normal BMI increases above 25 in old age, as previous studies have found that above normal BMI in old age might be associated with better functional capacity and reduced physical and cognitive decline [64]. In one of our earlier studies, old adults (85+) with obesity (opposed to overweight) had more considerable activity limitations than those with normal weight [65].

As the study population was stratified by gender, it is easier to show different factors associated with men’s and women’s MPI-leisure total score as well as each activity participation. For example, insomnia negatively affected both men and women visiting friends, men visiting relatives, and women going on an excursion. This aspect of psychological distress seems to affect women’s physical capacity and men’s social capacity. Regarding taking a ride in a car, none of the selected variables were related to women’s participation. For men, depression was negatively associated with this activity. Furthermore, fewer women than men had a car, suggesting the need and interest in driving differed between men and women. This finding was somewhat expected and in accordance with the notion that women used bikes and public transportation more often than men [66]. Among the participants who had cars in this study population, we found that women were not affected by the psychological distress when/if they consider taking a ride in a car. Whether this is due to their necessity to use cars, or higher resiliency among the women compared to men is not known. Future research may explore more factors that influence older men and women’s concern to ride in a car and their purpose to keep driving in old age. Women’s preferences even in modern urban areas can be explained by “typical” gender tasks due to their social roles of any age [66].

Pain severity/intensity and multimorbidity showed insignificance in association with activity participation in this target population. Their direct correlations to MPI-leisure total score were also very low. As mentioned before, most previous studies focused on relationships between pain, comorbidities, and physical activity since physical health and physical disability were major consequences of pain and multimorbidity [28,67]. In the current study population, the vast majority of the older adults were free from pain or only mild pain (77%) and only 4% suffered severe pain. Pain that threatens to limit activity was found in only a few people. About 5% of the women rated their pain intensity up to severe pain, and having severe pain seemed to motivate them to visit friends more than others without pain or with mild pain. All the psychological distress factors (i.e., depression, insomnia severity, and pain catastrophising) showed negative effects on visiting friends for women. Taken together, we might interpret that women with severe pain may present a greater need to maintain a social network despite pain and pain-related psychological distress. This finding was not unexpected, and it can be partially explained by the fact that participation increases life satisfaction and happiness in older adults irrespective of their symptoms [10,14,16].

Multimorbidity, on the other hand, was negligible in the regression models for the whole study sample and for gender stratification. We also tested higher cut-offs (*n* = 3, 4, or 5) for dichotomisation of multimorbidity to examine whether higher comorbidity burdens would negatively affect activity participation. Moreover, instead of multimorbidity, we tested a number of comorbidities as ordinal variables in the models. None of these approaches showed significant contributions to the outcomes (data not shown). Thus, our results suggest that living with multimorbidity was common in old age, and multimorbidity per se did not limit activity participation for both men and women. This finding was somewhat unexpected. A study of older Europeans has shown that people with multimorbidity participated less frequently in leisure activities compared to people without multimorbidity [68]. However, regarding activities from a wider perspective, multimorbidity may not always be a barrier in old age [69]. Some aging studies have also highlighted the importance of shifting toward psychological well-being instead of looking for causality of ill health in a multifactorial aging process [70,71]. Therefore, the literature usually supports the promotion of physical activity for those with chronic conditions to prevent health complications derived from the association between physical inactivity, sedentary behaviour, and multimorbidity [72,73].

Our findings may remind health professionals to assess older adults’ activity participation from a bio-psycho-social perspective. Having several morbidities, which is very common in old age, may not always be blamed as a cause of being inactive. Being overweight but not obese, is acceptable to keep an active aging life. On the other hand, not pain condition per se, but possibly the important roles of psychological consequences should be addressed. To prevent, identify, and adequately treat depressive symptoms or sleep disorders may positively affect older adults’ activity participation. Heath care providers should consider that decreased distress levels might foster leisure activity participation in older adults. Our results suggest that promoting participation involvement by targeting on interventions dealing with psychological distress may be an effective strategy for healthy mental aging regardless of chronic pain conditions. Other factors, such as gender aspects and culture interactions are also necessary to be taken into account. There is a need to deliver health services that align with older adults’ priority of maintaining active as they age.

This study has several limitations. First, our results, which are based on a large cross-sectional study design, cannot draw any causal conclusions. Therefore, we cannot explore older adults’ changes in behaviour during the aging process. Although the cross-sectional analysis could not identify whether the associations preceded or followed the increase of BMI or onset of psychological symptoms, our findings provide an insight into the impact of these modifiable factors on activity participation. Second, we did not examine the roles of social networks and social support, which may affect individual choices of leisure activities. Some inevitable incidents, such as retirement and loss of friends and relatives, reduce the social networks of older adults. How the consequences (i.e., feelings of loneliness) [74] influence older adults’ participation was not examined. Third, a simple calculation to define comorbidity and multimorbidity by adding the number of diagnoses made an easy approach in quantitative analysis but reduced the information on disease severity related to functional disability and activity limitation. Finally, the generalisability of the results does not cover older adults with severe cognitive impairment. This epidemiological study, which collected data using surveys, mainly focuses on chronic pain and health in the elderly. Questionnaires that measured experiences of chronic pain and its consequences require the respondents to have a certain cognitive capacity. For older adults with severe cognitive impairment, alternatives to pain measurement and interviews with their caregivers and relatives should be considered. This suggests that future studies might need to add these strategies in their research plan for this special patient group.

## 5. Conclusions

In conclusion, we found that older men and older women had different levels of participation in leisure activities. Activity participation was also related to BMI status and psychological distress but was not affected by pain severity or multimorbidity. To increase older men’s and women’s participation in leisure activities, health professionals and social workers should consider gender characteristics and target the potentially modifiable factors such as weight status and psychological distress.

## Figures and Tables

**Table 1 ijerph-18-02795-t001:** Characteristics of the study population and stratified by gender.

Variables	Characteristics	All Responders (*n* = 6611)	Men (*n* = 3057)	Women (*n* = 3554)	*p*-Value
Age, years, Mean (SD)		76.2 (7.4)	75.7 (7.1)	76.6 (7.6)	<0.001
Marital status	Currently married	3769 (57)	2106 (68.9)	1663 (46.8)	<0.001
Education level ^1^	Compulsory school	3335 (52.3)	1316 (44.3)	2019 (59.3)	<0.001
	Upper secondary school	1647 (25.8)	949 (31.9)	698 (20.5)	
	College/University	1394 (21.9)	709 (23.8)	685 (20.1)	
Income level (SEK/year)	<150,000	1974 (29.9)	287 (9.4)	1687 (47.5)	<0.001
	150,001~220,000	2262 (34.2)	1131 (37)	1131 (31.8)	
	>220,000	2375 (35.9)	1639 (53.6)	736 (20.7)	
Pain NRS 0–10	Without pain or mild (0–4)	5077 (77)	2499 (82)	2578 (73)	<0.001
	Moderate (5–7)	1245 (19)	455 (15)	790 (22)	
	Severe pain (8–10)	289 (4)	103 (3)	186 (5)	
Body mass index (BMI)	Normal	2836 (45)	1251 (43)	1585 (47)	<0.001
	Underweight	102 (2)	12 (0.5)	90 (3)	
	Overweight	2434 (39)	1269 (43)	1165 (35)	
	Obese	871 (14)	390 (13.5)	481 (15)	
Depression		4.7 (3.6)	4.3 (3.4)	5.0 (3.8)	<0.001
Anxiety		4.9 (4.5)	4.3 (4.2)	5.4 (4.7)	<0.001
Insomnia Severity		9.8 (5.5)	9.4 (5.5)	10.1 (5.5)	0.003
Pain Catastrophizing		11.9 (9.6)	10.9 (9.2)	12.9 (9.8)	<0.001
Multimorbidity	None or Single morbidity (0–1)	3091 (47)	1500 (49)	1591 (45)	<0.001
	Multimorbidity (2–12)	3520 (53)	1557 (51)	1963 (55)	
Participation aspects	Total score	2.4 (1.1)	2.4 (1.0)	2.5 (1.1)	<0.001
	Playing cards or other games	1.4 (1.7)	1.4 (1.6)	1.5 (1.7)	0.003
	Visiting friends	3.1 (1.5)	2.9 (1.4)	3.1 (1.5)	<0.001
	Take a ride in a car ^2,3^	1.6 (1.7)	1.7 (1.7)	1.5 (1.7)	0.005
	Visiting relatives ^4,5^	3.2 (1.5)	3.0 (1.5)	3.4 (1.6)	<0.001
	Make an excursion	2.8 (1.5)	2.8 (1.5)	2.8 (1.6)	0.567

Missing data: ^1^ = 235, ^2^ = 306 (men without car), ^3^ = 1038 (women without car), ^4^ = 336 (men without relatives within 150 km), ^5^ = 402 (women without relatives within 150 km). SEK: Swedish crowns, 1 SEK ≈ 0.1 Euro.

**Table 2 ijerph-18-02795-t002:** Pearson correlations of participation, psychological distress, BMI, comorbidities, pain intensity, and age for the whole sample and stratified by gender.

Variables	MPI-Leisure Total Score		
	All	Men	Women
Age	−0.20 ***	−0.12 ***	−0.27 ***
Currently married (vs. not married)	−0.09 **	−0.09 **	−0.11 **
Education level (low to high level)	0.02	−0.01	0.06 **
Yearly income level (low to high level)	0.07 **	0.09 **	0.12 **
Pain Intensity	−0.03 **	−0.03	−0.05 **
BMI	0.08 ***	0.09 ***	0.07 ***
Number of comorbidities	−0.10 ***	−0.07 ***	−0.12 ***
Depression	−0.29 ***	−0.25 ***	−0.27 ***
Anxiety	−0.16 ***	−0.14 ***	−0.20 ***
Insomnia severity	−0.14 ***	−0.12 **	−0.16 ***
Pain catastrophizing	−0.06 ***	−0.03	−0.11 ***

** *p* < 0.01, *** *p* < 0.001.

**Table 3 ijerph-18-02795-t003:** Results of multivariable regression analysis on total participation and participation aspects for the total sample adjusted for age, marriage status, education level, and yearly income level.

All (*N* = 6611)		Participation Aspects				
Variables	Total Score	Playing Cards or Other Games	Visiting Friends	Taking a Ride in a Car ^1^	Visiting Relatives ^2^	Making an Excursion
	B (95% CI)	B (95% CI)	B (95% CI)	B (95% CI)	B (95% CI)	B (95% CI)
**Gender (reference: men)**						
Women	**0.25 (0.14, 0.37) *****	0.05 (−0.14, 0.25) ^ns^	**0.39 (0.23, 0.55) *****	−0.08 (−0.28, 0.13)^ns^	**0.47 (0.29, 0.66) *****	**0.28 (0.12, 0.44) ****
**Pain (reference: none or mild pain)**						
Moderate pain	0.03 (−0.09, 0.15) ^ns^	0.06 (−0.14, 0.26) ^ns^	0.07 (−0.09, 0.23) ^ns^	0.04 (−0.18, 0.26) ^ns^	−0.04 (−0.23, 0.15) ^ns^	0.04 (−0.13, 0.21) ^ns^
Severe pain	0.21 (−0.03, 0.45) ^ns^	0.08(−0.31, 0.47) ^ns^	0.30 (−0.02, 0.62) ^ns^	0.23 (−0.23, 0.68) ^ns^	0.16 (−0.21, 0.53) ^ns^	0.19 (−0.13, 0.52) ^ns^
**BMI (reference: normal weight)**						
Underweight	−0.14 (−0.51, 0.22) ^ns^	0.08 (−0.51, 0.66) ^ns^	−0.20 (−0.68, 0.28) ^ns^	−0.23 (−0.92, 0.47) ^ns^	−0.11 (−0.71, 0.48) ^ns^	−0.28 (−0.78, 0.22) ^ns^
Overweight	**0.22 (0.11, 0.33) *****	**0.20 (0.02, 0.38) ***	**0.16 (0.01, 0.31) ***	**0.42 (0.22, 0.61) *****	**0.21 (0.04, 0.39) ***	**0.16 (0.01, 0.31) ***
Obese	0.03 (−0.13, 0.19) ^ns^	**0.28 (0.02, 0.53) ***	−0.09 (−0.30, 0.12) ^ns^	**0.32 (0.04, 0.60) ***	0.01 (−0.23, 0.26) ^ns^	**−0.23 (−0.45, −0.01) ***
**Comorbidities (reference: none or single morbidity)**						
Multimorbidity	0.07 (−0.05, 0.18) ^ns^	0.10 (−0.09, 0.28) ^ns^	0.02 (−0.13, 0.17) ^ns^	−0.08 (−0.28, 0.13) ^ns^	0.12 (−0.06, 0.30) ^ns^	−0.07 (−0.23, 0.08) ^ns^
**Psychological distress**						
Depression	**−0.61 (−0.81, −0.40) *****	**−0.48 (−0.80, −0.15) ****	**−0.83 (−1.10, −0.56) *****	**−0.57 (−0.94, −0.20) ****	**−0.71 (−1.02, −0.4) *****	**−0.47 (−0.75, −0.19) ****
Insomnia Severity	**−0.02 (−0.03, −0.01) ****	0.01 (−0.01, 0.02)^ns^	**−0.03 (−0.05, −0.02) *****	−0.02 (−0.03, 0.01)^ns^	**−0.02 (−0.03, −0.01) ***	**−0.03 (−0.04, −0.01) *****
Pain Catastrophizing	−0.01 (−0.01, 0.01) ^ns^	−0.01 (−0.01, 0.01) ^ns^	−0.01 (−0.01, 0.01) ^ns^	−0.01 (−0.01, 0.01) ^ns^	−0.01 (−0.02, 0.01) ^ns^	**−0.01 (−0.02, −0.01) ***

CI: confidence interval, ns = non-significant; * *p* < 0.05, ** *p* < 0.01, *** *p* < 0.001. All significant values are given in bold. Ref: reference values. Missing data: ^1^ = 1344 (without car), ^2^ = 738 (without relatives within 150 km).

**Table 4 ijerph-18-02795-t004:** Results of multivariable regression analysis on total participation and participation aspects stratified by gender adjusted for age, marriage status, education level, and yearly income level.

Participation Aspects						
Variables	Total Score	Playing Cards or Other Games	Visiting Friends	Taking a Ride in a Car ^1,2^	Visiting Relatives ^3,4^	Making an Excursion
	B (95% CI)	B (95% CI)	B (95% CI)	B (95% CI)	B (95% CI)	B (95% CI)
**Men (*n* = 3057)**						
**Pain (reference: none or mild pain)**						
Moderate pain	0.13 (−0.06, 0.33) ^ns^	0.23 (−0.09, 0.56) ^ns^	0.10 (−0.15, 0.35) ^ns^	0.10 (−0.23, 0.43) ^ns^	0.20 (−0.11, 0.52) ^ns^	0.08 (−0.19, 0.36) ^ns^
Severe pain	0.17 (−0.22, 0.57) ^ns^	0.27 (−0.39, 0.93) ^ns^	−0.08 (−0.59, 0.43) ^ns^	0.17 (−0.56, 0.89) ^ns^	0.21 (−0.40, 0.82) ^ns^	0.12 (−0.43, 0.68) ^ns^
**BMI (reference: normal weight)**						
Underweight	−0.76 (−1.89, 0.37) ^ns^	−1.22 (−3.11, 0.67) ^ns^	−1.21 (−2.67, 0.25) ^ns^	−0.86 (−2.71, 0.98) ^ns^	0.57 (−1.52, 2.65) ^ns^	−0.28 (−1.86, 1.31) ^ns^
Overweight	**0.26 (0.09, 0.43) ****	**0.30 (0.01, 0.58) ***	0.11 (−0.11, 0.32) ^ns^	**0.59 (0.31, 0.88) *****	0.14 (−0.13, 0.41) ^ns^	0.20 (−0.04, 0.43) ^ns^
Obese	−0.01 (−0.24, 0.25) ^ns^	0.33 (−0.08, 0.74) ^ns^	0.04 (−0.28, 0.35) ^ns^	0.27 (−0.17, 0.71) ^ns^	−0.08 (−0.47, 0.31) ^ns^	**−0.40 (−0.74, −0.05) ***
**Comorbidities (reference: none or single morbidity)**						
Multimorbidity	0.07 (−0.10, 0.24) ^ns^	0.11 (−0.17, 0.40) ^ns^	0.08 (−0.14, 0.30) ^ns^	0.12 (−0.17, 0.42) ^ns^	0.06 (−0.22, 0.33) ^ns^	−0.09 (−0.33, 0.15) ^ns^
**Psychological distress**						
Depression	**−0.61 (−0.94, −0.28) *****	−0.38 (−0.93, 0.17) ^ns^	**−0.84 (−1.27, −0.42) *****	**−0.72 (−1.30, −0.15) ***	**−0.60 (−1.11, −0.09) ***	**−0.54 (−1.01, −0.08) ***
Insomnia Severity	−0.02 (−0.03, 0.01) ^ns^	−0.01 (−0.03, 0.02) ^ns^	**−0.04 (−0.06, −0.02) *****	−0.01 (−0.03, 0.02) ^ns^	−0.03 (−0.02, −0.01) *	−0.01 (−0.04, 0.01) ^s^
Pain Catastrophizing	−0.01 (−0.01, 0.01) ^ns^	−0.01 (−0.02, 0.01) ^ns^	0.01 (−0.01, 0.02) ^ns^	−0.01 (−0.02, 0.01) ^ns^	−0.01 (−0.02, 0.01) ^ns^	−0.01 (−0.02, 0.01) ^ns^
**Women (*n* = 3554)**						
**Pain (reference: none or mild pain)**						
Moderate pain	−0.04 (−0.20, 0.11) ^ns^	−0.04 (−0.29, 0.21) ^ns^	0.05 (−0.16, 0.25) ^ns^	0.25 (−0.33, 0.84) ^ns^	−0.19 (−0.44, 0.05) ^ns^	0.01 (−0.21, 0.21) ^ns^
Severe pain	−0.21 (−0.09, 0.51) ^ns^	−0.06 (−0.53, 0.42) ^ns^	**0.49 (0.09, 0.89) ***	−0.02 (−0.30, 0.27) ^ns^	0.13 (−0.34, 0.60) ^ns^	0.22 (−0.19, 0.62) ^ns^
**BMI (reference: normal weight)**						
Underweight	−0.08 (−0.47, 0.31) ^ns^	0.16 (−0.46, 0.78) ^ns^	−0.06 (−0.58, 0.46) ^ns^	−0.16 (−0.92, 0.59) ^ns^	−0.12 (−0.75, 0.51) ^ns^	−0.28 (−0.81, 0.25) ^ns^
Overweight	**0.18 (0.04, 0.33) ***	0.12 (−0.12, 0.36) ^ns^	**0.21 (0.01, 0.41) ***	0.26 (−0.01, 0.53) ^ns^	**0.27 (0.04, 0.50) ***	0.11 (−0.09, 0.32) ^ns^
Obese	0.04 (−0.16, 0.25) ^ns^	0.24 (−0.10, 0.56) ^ns^	−0.18 (−0.46, 0.09) ^ns^	0.35 (−0.03, 0.72) ^ns^	0.09 (−0.23, 0.41) ^ns^	−0.13 (−0.40, 0.15) ^ns^
**Comorbidities (reference: none or single morbidity)**						
Multimorbidity	0.06(−0.09, 0.21) ^ns^	0.09 (−0.15, 0.32) ^ns^	−0.03 (−0.23, 0.17) ^ns^	0.08 (−0.19, 0.35) ^ns^	0.13 (−0.10, 0.37) ^ns^	−0.08 (−0.30, 0.13) ^ns^
**Psychological distress**						
Depression	**−0.60 (−0.86, −0.34) *****	**−0.53 (−0.94, −0.13) ****	**−0.81 (−1.15, −0.46) *****	−0.42 (−0.90, 0.07) ^ns^	**−0.81 (−1.20, −0.42) *****	**−0.41 (−0.76, −0.06) ****
Insomnia Severity	**−0.02 (−0.03, −0.01) ***	−0.02 (−0.01, 0.04) ^ns^	**−0.03 (−0.05, −0.01) ****	−0.02 (−0.04, 0.01) ^ns^	−0.01 (−0.03, 0.01) ^ns^	**−0.03 (−0.05, −0.02) *****
Pain Catastrophizing	−0.01 (−0.02, 0.01) ^ns^	−0.01 (−0.02, 0.01) ^ns^	**−0.02 (−0.03, −0.01) ****	−0.01 (−0.02, 0.01) ^ns^	−0.01 (−0.02, 0.01) ^ns^	−0.01 (−0.02, 0.01) ^ns^

CI: confidence interval, ns = non-significant; * *p* < 0.05, ** *p* < 0.01, *** *p* < 0.001. All significant values are given in bold. Ref: reference values.Missing data: ^1^ = 306 (men without car), ^2^ = 1038 (women without car), ^3^ = 336 (men without relatives within 150 km), ^4^ = 402 (women without relatives within 150 km).

## Data Availability

Data are available from the corresponding author on reasonable request.

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
