# Peer review of "Association between Participation Activities, Pain Severity, and Psychological Distress in Old Age: A Population-Based Study of Swedish Older Adults"

_ijerph, 2021, doi:10.3390/ijerph18062795_

Round 1

Reviewer 1 Report

An excellent paper, and a pleasure to read! 

Excellent statistical analysis!

Some comments follow: 

Page4 lines 191 to 192: Please provide more detail about education level and income level of participants, for example education levels, income levels, also if there are any significant differences in these between women and men.  What is provided in this sentence is not particularly helpful. 

page 5 lines 196 to 197 - weight - fraction of men and women who were overweight or obese?  

page 11 lines 329-330.  What does this mean??  Please revise this sentence to more adequately summarize the differences you found between men and women.  I would especially consider and comment on the observations that , while you mentioned that women had higher psychological morbidities, and that this was linked to decreased activities, you also found that women engaged in more activities.  Are there factors here, maybe increased connection,  or something else perhaps, that may contribute to increased resiliency in the population of women?  Or maybe just asking the question as a further conjecture for future study?  

Thanks, and good luck. 

Author Response

We thank the reviewer for all the very insightful comments and useful directions. All changes are underlined throughout the text. 

Reviewer nr 1

An excellent paper, and a pleasure to read! 

Excellent statistical analysis!

  • We thank the reviewer.

Some comments follow: 

-Page4 lines 191 to 192: Please provide more detail about education level and income level of participants, for example education levels, income levels, also if there are any significant differences in these between women and men.  What is provided in this sentence is not particularly helpful. 

  • Our comments: Thank you for your suggestion. We classified the educational level into the following three categories: compulsory school (elementary/secondary), upper secondary school (or vocational training), and university/college. We added this part in our method, please see p.3, lines 89-91. The total annual income was assessed according to the SCB and was divided into three levels as provided in the text (p3, lines 94-95).

In the results part, we gave more detailed information about gender differences. We have added that “A lower proportion of women (20.1%) than men (23.8%) earned university or college education (P<0.001). Over one in three participants (35.9%) were classified into the highest income level (>220 000 Swedish Crowns/year, 2012). More than half of men (53.6%), whereas only one-fifth women (20.7) had highest income level (P<0.001). Most women’s incomes (47.5%) were classified into the lowest level (<150 000 Swedish Crowns/year), and there was only a minority of men (9.4%) in this level (P<0.001)”, p5, lines 194-200.

-page 5 lines 196 to 197 - weight - fraction of men and women who were overweight or obese?  

  • Our comments: Thank you for your suggestion. In the revised paper, we added more information in the results part, please see p5, line 204-205: There were more men than women in the overweight category (43% vs 35%). A slightly higher proportion of women than men were obese (15% vs 13.5%).

-page 11 lines 329-330.  What does this mean??  Please revise this sentence to more adequately summarize the differences you found between men and women.  I would especially consider and comment on the observations that , while you mentioned that women had higher psychological morbidities, and that this was linked to decreased activities, you also found that women engaged in more activities.  Are there factors here, maybe increased connection,  or something else perhaps, that may contribute to increased resiliency in the population of women?  Or maybe just asking the question as a further conjecture for future study?  

  • Our comments: Thank you for your concern. In the revised paper, we have made our expression clearer and connected to our findings. Thus, we rewrote the sentence, please see p11, line 337-342: Among the participants who had cars in this study population, we found that women were not affected by the psychological distress when/if they consider taking a ride in a car. Whether this is due to their necessity to use cars, or higher resiliency among the women compared to men is not known. Future research may explore more factors that influence older men and women’s concern to ride in a car and their purpose to keep driving in old age.        

Reviewer 2 Report

Dear researchers and authors of this article,

thank you for sending me this interesting manuscript to review.

Overall, this is an interesting article providing information on an important aspect of participation activities, pain severity, and psychological distress in the elderly. The manuscript is well written, and the data analytical strategy is also appropriate. The large sample size of the participants and the stratified sampling method give merit to the cross-sectional design.

However, some remarks related to the discussion section should be further clarified, developed, and explained.

Abstract: no remarks

Introduction: it was well written and fitted to the context

Methods and results: appropriate data analysis and presentation. The description of the stratified sampling method is adequately described. However, please give an example of how the data collection evolved in your study during the postal reminders (p. 2, line 91-92).

Discussion: Overall the discussion reads very well. However, some points need clarification.

The discussion should discuss what the findings imply for the Swedish elderly and general for the elderly.

Clinical implications should be also underlined.

The authors found that gender, body mass index (BMI) levels, and psychological distress factors significantly affected older adults’ participation in leisure activities. A clarification of how these results should be useful to the older adults in terms of recommendations would be a merit.  Please also try to give more explanations to your findings.

Author Response

We thank the reviewer for all the very insightful comments and useful directions. All changes (including the comments from reviewer nr 1) are underlined throughout the text. 

Reviewer nr 2

thank you for sending me this interesting manuscript to review.

Overall, this is an interesting article providing information on an important aspect of participation activities, pain severity, and psychological distress in the elderly. The manuscript is well written, and the data analytical strategy is also appropriate. The large sample size of the participants and the stratified sampling method give merit to the cross-sectional design.

However, some remarks related to the discussion section should be further clarified, developed, and explained.

  • We thank the reviewer.

Abstract: no remarks

Introduction: it was well written and fitted to the context

Methods and results: appropriate data analysis and presentation. The description of the stratified sampling method is adequately described. However, please give an example of how the data collection evolved in your study during the postal reminders (p. 2, line 91-92).

  • Our comments: Thank you for your suggestion. The summary of the data collection is listed as below:

                                             N                       %

After 1st invitation letter 5 543                55,7

After 1st reminder letter  829                   8,3

After 2nd reminder letter 320                   3,2

Non responders                 3 261                32,8

Sum                                     9 953                100,0

Over coverage                         47

We believe that the two postal reminders were useful to reduce nonresponse rate. Except for 47 persons without registered address in this target population, most responses (5543, 55.7%) were given after the first invitation letters were sent. In the revised paper, we have now added more relevant information in the text, please see p4, line188-189: More than 1100 responses (11.5%) were received after two postal reminders.

Discussion: Overall the discussion reads very well. However, some points need clarification. The discussion should discuss what the findings imply for the Swedish elderly and general for the elderly.

  • Our comments: Thank you for indicating this point. As per answer below we have added some recommendations based on our findings that would apply not only to Swedish older adults but also in older adults in general, please see p12, lines 385-390: To prevent, identify and adequately treat depressive symptoms or sleep disorders, may positively affect older adults’ activity participation. Heath care providers should consider that decreased distress levels might foster leisure activity participation in older adults. Our results suggest that promoting participation involvement by targeting on interventions dealing with psychological distress may be an effective strategy for healthy mental aging regardless of chronic pain conditions.

Clinical implications should be also underlined.

  • Our comments: Thank you for your suggestion. We agree that it is a very important aspect to enhance the findings in connection to the ‘real-life’ clinical practice. In the revised paper, we tried to develop this aspect which we shortly mentioned at the beginning of the discussion (our results suggest prevention and intervention approaches to these potentially modifiable factors should improve activity participation in older adults.), please see p12, lines 380-394:
    Our findings may remind health professionals to assess older adults’ activity participation from a bio-psycho-social perspective. Having several morbidities, which is very common at the old age, may not always be blamed as a cause of being inactive. Being overweight but not obese, is acceptable to keep an active aging life. On the other hand, not pain condition per se, but possibly the important roles of psychological consequences should be addressed. To prevent, identify and adequately treat depressive symptoms or sleep disorders, may positively affect older adults’ activity participation. Heath care providers should consider that decreased distress levels might foster leisure activity participation in older adults. Our results suggest that promoting participation involvement by targeting on interventions dealing with psychological distress may be an effective strategy for healthy mental aging regardless of chronic pain conditions. Other factors, such as gender aspects and culture interactions are also necessary to be taken into consideration in the assessment. There is a need to deliver health services that align with older adults’ priority of maintaining active as they age.

The authors found that gender, body mass index (BMI) levels, and psychological distress factors significantly affected older adults’ participation in leisure activities. A clarification of how these results should be useful to the older adults in terms of recommendations would be a merit. Please also try to give more explanations to your findings.

  • Our comments: Thank you. It is very helpful with your suggestion. We tried to consider this suggestion when we formulated the clinical implication. In general, recommendations can be difficult since our findings were based on the cross-sectional analysis and no causal conclusions could be drawn. However, we noted in the clinical implications, for example, pain conditions, overweight status, multimorbidity are acceptable and may not contribute to the limitations of men and women’s participation. In the revised paper, as per previous comment we have added in the text: Heath care providers should consider that decreased distress levels might foster leisure activity participation in older adults. Our results suggest that promoting participation involvement by targeting on interventions dealing with psychological distress may be an effective strategy for healthy mental aging regardless of chronic pain conditions. Please see p12, lines 386-390.
